

# Acute effects of optimal power load flywheel half-squat training on lower limb explosive power under different load volumes

Haonan Qi[1,*], Mushuai Hao[1,*], Boyang Qu[2], Liang Zhao[3] and Wei Han[3]

[1] School of Graduate Education, Shandong Sport University, Jinan, Shandong Province, China
[2] Shandong Sports Science Research Center, Jinan, Shandong Province, China
[3] School of Competitive Sport, Shandong Sport University, Rizhao, Shandong Province, China
* These authors contributed equally to this work.

## ABSTRACT

**Background:** To explore the effects of flywheel half-squat interventions with different volumes of optimal power load (OPL) on post-activation performance enhancement (PAPE) in countermovement jump (CMJ) height and 30 m sprint performance among collegiate athletes.

**Methods:** A randomized crossover design was employed, recruiting 20 collegiate athletes to participate in the experiment. After determining each participant's OPL, four different training load schemes were arranged for eight formal experiments, including four CMJ tests and four 30 m sprint tests. The differences between baseline and post-intervention at 0, 4, 8, and 12 min were compared. A two-factor repeated measures ANOVA was used for data analysis, with a significance level set at $P < 0.05$, and Cohen's d value was used to represent the effect size (ES).

**Results:** (1) The improvement in CMJ height for different flywheel half-squat load volumes peaked at 8 min of recovery. In terms of ES, the improvement was Group B > Group C > Group A. (2) The improvement in 30 m sprint speed for Group A peaked at 4 min post-intervention, while the improvement for Groups B and C peaked at 8 min post-intervention. In terms of ES, the improvement was Group B > Group C > Group A.

**Conclusion:** Using two sets × six repetitions of OPL flywheel half-squat arrangement can induce a more reliable PAPE effect compared to higher load volumes. However, when using half-squats as a pre-stimulation exercise, the PAPE effect on CMJ height is superior to that on 30 m sprint speed.

## INTRODUCTION

Post-activation performance enhancement (PAPE) refers to a physiological phenomenon where voluntary explosive power output is temporarily enhanced following high intensity resistance contractions under specific conditions (*Blazevich & Babault, 2019*;

Corresponding authors
Liang Zhao,
zhaoliang1@sdpei.edu.cn
Wei Han, hanwei@sdpei.edu.cn

*Cuenca-Fernández et al., 2017*). Its physiological mechanisms may encompass an elevation in muscle temperature, an augmentation of water content within the muscle, as well as other mechanisms that remain to be elucidated (*Blazevich & Babault, 2019*). When the human body is stimulated by external load, fatigue and enhancement (neuromuscular efficiency and motor unit recruitment rate increase) effects will occur at the same time. With the extension of rest time, when the enhancement effect is greater than the fatigue effect, the muscle contraction performance will be improved, and PAPE will occur (*Xenofondos et al., 2018*).

Despite some controversy surrounding its physiological mechanisms, PAPE as a result of warm-up, can enhance subsequent exercise performance. In traditional resistance training, using high-intensity loads (85–90% 1RM) can induce significant PAPE (*Garbisu-Hualde & Santos-Concejero, 2021*). However, traditional resistance equipment has limitations such as heavy weight, inconvenience in transport, and minimal eccentric phase stimulation (*Petré, Wernstål & Mattsson, 2018*; *Handford et al., 2022*; *Duchateau & Enoka, 2016*). Therefore, considering these limitations, flywheel training devices have garnered attention and have been applied in PAPE research (*Petré, Wernstål & Mattsson, 2018*; *de Keijzer, Gonzalez & Beato, 2022*).

Initially, flywheel training devices were designed to help astronauts maintain muscle mass and prevent atrophy in the weightless environment of space (*Buonsenso et al., 2023*; *Berg & Tesch, 1994*). Current research has found that flywheels offer significant advantages in enhancing athletic performance and preventing injuries (*de Keijzer, Gonzalez & Beato, 2022*; *Li et al., 2023*; *Beato et al., 2021a*; *Sagelv et al., 2020*; *de Hoyo et al., 2015a*; *Su, 2021*). This training method provides maximum resistance throughout the entire range of motion, particularly offering greater resistance during the eccentric phase than the concentric phase, thereby achieving eccentric overload (EO) (*Petré, Wernstål & Mattsson, 2018*; *de Keijzer, Gonzalez & Beato, 2022*; *Weng & Wei, 2021*; *Norrbrand, Pozzo & Tesch, 2010*; *Norrbrand et al., 2008*). Studies have shown that increasing eccentric load may lead to higher activation of type II muscle fibers (*Maroto-Izquierdo et al., 2017*). It is important to note that flywheel equipment cannot use traditional 1RM metrics to quantify load intensity. In current flywheel training studies, researchers often use predetermined inertial loads to monitor and adjust training intensity, ensuring all subjects use a consistent inertia (*Beato et al., 2021b*, *2021c*). However, due to the inherent dependence of flywheel inertial training on concentric output, controlling training intensity based solely on adjusting inertia may not accurately align with specific training plans that require prescribed loads (*Carroll et al., 2019*). Therefore, some studies suggest that the built-in rotary encoder devices of flywheel trainers can record the peak power generated during flywheel training to monitor load intensity (*de Keijzer, McErlain-Naylor & Beato, 2022*; *Spudić, Smajla & Šarabon, 2020*). Generally, the optimal power load (OPL) for each practitioner can be determined through progressive measurement.

OPL refers to the external load at which skeletal muscle generates maximum power output during contraction (*Loturco et al., 2017*; *Liang et al., 2021*). Strength training at OPL allows athletes to enhance their maximal power output while minimizing fatigue, thereby translating training gains more effectively into athletic performance (*Sarabia et al.,*

2017; *Loturco et al., 2014*). To date, studies have tested the reliability of using OPL as an intensity measure for flywheel training (*de Hoyo et al., 2015a, 2015b; Maroto-Izquierdo, Bautista & Martín Rivera, 2020*). However, in current research on flywheel training-induced PAPE, studies using OPL as the intensity measure have employed a single load volume. For instance, *de Hoyo et al. (2015b)* used a training scheme of 4 sets × 6 repetitions, while *Maroto-Izquierdo, Bautista & Martín Rivera (2020)* used a scheme of 1 set × 6 repetitions. Only one study has investigated the effects of flywheel training with different load volumes on jump ability PAPE, however it did not apply individualized load intensities, using only a single load intensity throughout (0.029 kg·m$^2$) (*de Keijzer et al., 2020*). It has been shown that PAPE is influenced by load volumes, which significantly affects the final PAPE induction effect (*Garbisu-Hualde & Santos-Concejero, 2021*). Therefore, this study aims to explore the effects of flywheel half-squat training with different load volumes on lower limb explosive power PAPE among collegiate athletes, based on individualized training loads (OPL), to provide more practical basis for flywheel training to induce PAPE.

## METHODS

### Study subjects

This study aims to explore the acute effects of individualized inertial intensities with varying load volumes during flywheel half-squat training on lower limb explosive power. Collegiate athletes were recruited for the experiment.

### Research methods

#### Experimental subjects

Twenty male athletes with national level 2 or above qualifications were recruited from the Rizhao Campus of Shandong Sport University (sports items include basketball, handball and volleyball). Inclusion criteria were no lower limb injuries in the past 6 months and no other diseases. All subjects had at least 3 years of strength training experience. This study was approved by the Ethics Committee of Sports Science at Shandong Sport University (Approval No.: 2022030) and complied with the ethical requirements of the Helsinki Declaration and relevant Chinese laws and regulations (*de Hoyo et al., 2015a*). Subjects participated voluntarily with full knowledge of the experimental process and signed informed consent forms. To calculate the minimum sample size, statistical software (G*power, Dusseldorf, Germany) was used. In view of the two-factor repeated measures ANOVA used in the study, an a-error ≤0.05, a desired power (1-ß error) = 0.8, and an effect size (ES) $f$ = 0.38, this effect size was obtained from a meta-analysis (*Wilson et al., 2013*), resulting in a minimum sample size of 16 participants. To prevent any possible dropout from affecting the final statistical power, a total of 20 subjects were included in the study (Table 1).

#### Experimental equipment

Body composition analyzer (GAIA KIKO, Seoul, Korea), flywheel training device and supporting harness (kBox4, Stockholm, Sweden), flywheel built-in rotary encoder (kMeter,
| Table 1 Basic information of subjects (*n* = 20). | | | | | | | |
|---|---|---|---|---|---|---|---|
| Height (cm) | Weight (kg) | Age | Training years | Resistance training years | CMJ height (cm) | 30 m sprint performance (s) | Concentric peak power (W) |
| 180.4 ± 4.9 | 73.2 ± 6.9 | 20.1 ± 2 | 6.8 ± 2.5 | 4.9 ± 2 | 47.8 ± 4.41 | 4.19 ± 0.11 | 978.95 ± 243.98 |

Kristinehamn, Sweden), linear sensor (QueDong VBT pro, Shanghai, China), four-gate infrared velocity and sensitivity testing system (Brower Timing System, Draper, UT, USA), measuring tape.

### Experimental design

This study utilized a randomized crossover design, subjects were randomized before the formal experiment using EXCEL software. After selecting the subjects, all participants underwent training on how to use the flywheel training device, followed by OPL testing and a 1-week washout period to eliminate training and fatigue effects. After the washout period, all subjects completed a total of eight formal experiments, including four CMJ height tests and four 30 m sprint tests under different load volumes and no intervention. Each test included a baseline test and post-intervention tests (total of five tests). Referencing *Gouvêa et al. (2013)*, post-intervention tests were conducted at four time points: 0, 4, 8, and 12 min. The total duration for all experiments was 7 weeks. The subjects arranged 72 h intervals between the two experiments to eliminate the fatigue caused by each experiment.

## Experimental procedure and control
### Experimental procedure

Studies indicate that at least three familiarization sessions are required before the first use of the flywheel training device to ensure correct usage during training (*Bright et al., 2023*; *Beato et al., 2019a*). Therefore, 3 weeks before the formal experiment, subjects were briefed on the experimental procedure, test items, and precautions. Three familiarization sessions were arranged to provide technical guidance on flywheel half-squat exercises, ensuring subjects mastered the correct technique. In the familiarization course, subjects were instructed to complete the concentric muscle action of the half-squat as quickly as possible using the flywheel training device, during the eccentric phase, they aimed to achieve approximately 90° of knee flexion, light resistance was applied during the first third of the eccentric phase, and full resistance was applied during the remaining two-thirds, The goal was to stop the flywheel's rotation to achieve the optimal training effect (*de Keijzer et al., 2020*; *Bright et al., 2023*), before starting the next concentric movement. During the familiarization phase, researchers provided training protection for each subject to prevent injuries due to unfamiliarity with the training method.

After the three familiarization sessions, subjects underwent OPL testing. Before the test, a standardized warm-up was conducted, including 5 min of slow running followed by 5 min of dynamic preparation exercises. The dynamic exercises included bodyweight squats and movements targeting the hip, knee, and ankle joints. This warm-up protocol

was also applied in the formal experiments (*Beato et al., 2021c*). The testing procedure was based on the studies by *Maroto-Izquierdo et al. (2021)* and *Beato et al. (2021d)*, with an initial inertial load set at 0.025 kg·m$^2$. Each inertial load test consisted of three sets of six repetitions, with the first two repetitions of each set not counted to allow the flywheel to gain initial kinetic energy. This training arrangement was also used in the formal experiments. After each inertial load test, the average peak concentric power output of the last two sets of six repetitions was recorded, excluding the first set due to typically lower power output. The inertial load was gradually increased by 0.01 to 0.015 kg·m$^2$ until a decrease in power output was observed. The previous inertial load was then determined to be the subject's OPL. To avoid power decline due to fatigue, there was a 3-min rest interval between tests with the same inertial load and a 5-min rest interval between tests with different inertial loads. Power data was measured using the kMeter flywheel built-in rotary encoder. The tested OPL for the recruited subjects was 978.95 ± 243.98 W, with a corresponding inertia of 0.035 ± 0.01 kg·m$^2$.

Following the OPL test, a 1-week washout period was scheduled to eliminate training and fatigue effects before starting the formal experiments. Each experiment consisted of a warm-up, baseline test, flywheel half-squat intervention, and post-intervention tests at different recovery times.

Baseline tests included:

(1) CMJ test: The CMJ height was measured using a linear sensor (VBT pro), a method previously validated as an effective alternative to force platforms (*Cronin, Hing & McNair, 2004*; *Hojka et al., 2022*). During the test, one end of the linear sensor was securely attached to one side of the obstacle bar, which was then positioned across the back of the subject's neck, instructing them to grasp the obstacle bar with both hands. Subjects stood naturally with feet slightly wider than shoulder-width apart or shoulder-width apart. Upon hearing the researcher's "jump" command, subjects first bent their knees to squat (to a self-selected depth) and then jumped vertically with maximum effort. To minimize the interference of arm swing and horizontal body movement on the test results, subjects were instructed to grip the bar tightly and control horizontal displacement during the jump. If such errors occurred, the test was repeated. Each subject performed two jumps (after landing from the first jump, subjects stood still for 2–3 s before performing the second jump), and the best performance was recorded as the baseline for that experiment (Fig. 1).

(2) A total of 30 m sprint test: Before the test, infrared speed measurement devices were placed on a rubber track by the researchers, positioned at the starting line and the 30 m mark. After the warm-up, subjects stood at the designated starting position in a standing start posture as instructed. Upon hearing the command "run," they sprinted to the finish line as fast as possible. The infrared speed measurement devices started timing from the start line and recorded the 0–30 m sprint data when the subjects crossed the 30 m mark. Each subject performed two tests, with the best performance recorded as the baseline for that experiment. There was a 4-min rest interval between the two tests.

Following the baseline tests, the formal experimental intervention began. Each subject performed the training protocol at OPL intensity. Researchers provided the most suitable safety harness and adjusted the flywheel rope length according to each subject's height and

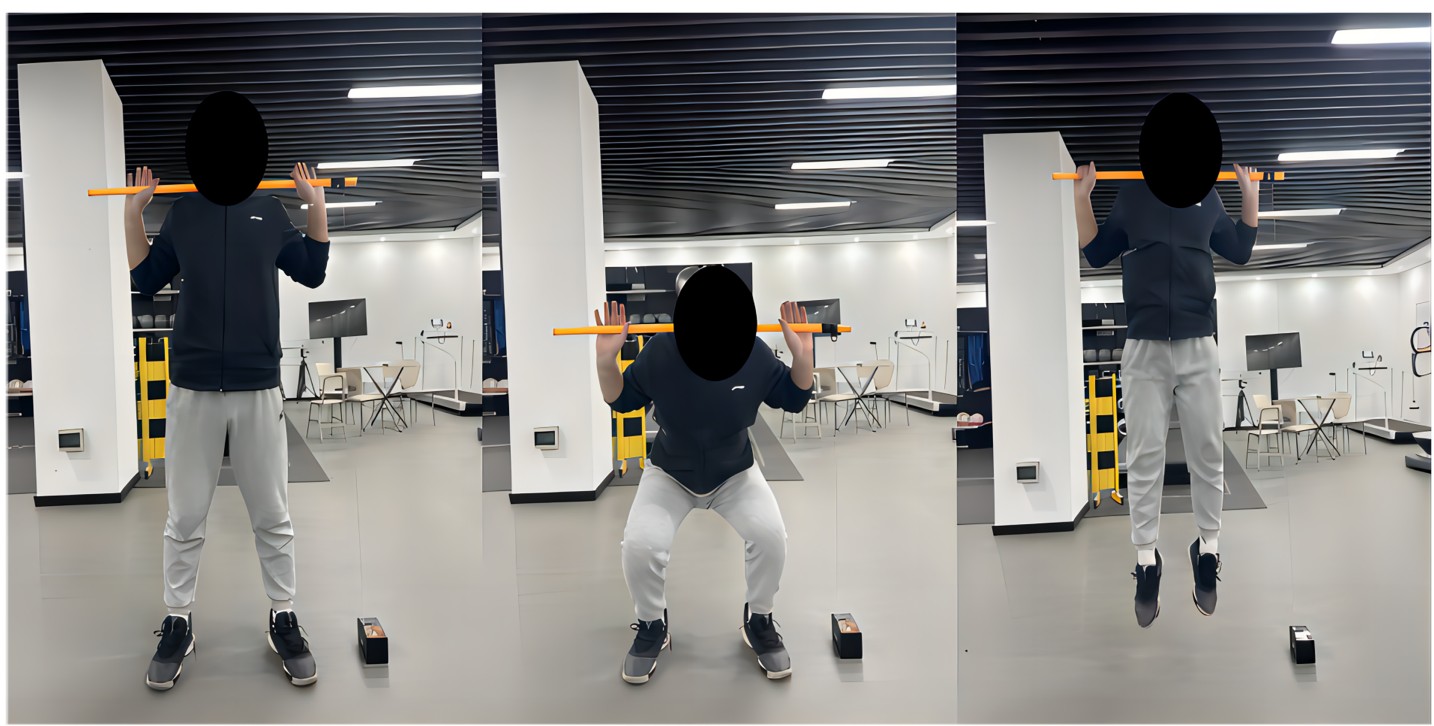

**Figure 1 CMJ height test illustrations.**

weight, offering verbal encouragement during the flywheel training. To avoid learning effects, subjects randomly completed one of the four intervention protocols, including the no-intervention control group (Table 2). For single-set exercises, there were no intervals, while for multiple-set interventions, there was a 2-min rest between sets, followed by post-intervention tests. At each time point, CMJ tests were conducted twice, with the best result used for data analysis, following the same standards as the baseline test. Due to time and fatigue considerations, the 30 m sprint test was conducted only once at each time point for data analysis, following the same standards as the baseline test. The experimental procedure is shown in Fig. 2.

## Experimental control

To minimize the influence of confounding factors on the experimental results, subjects were required to maintain a regular diet and sleep schedule during the experiment, avoid alcohol and caffeine, including all types of functional drinks, 24 h before the experiment. Subjects wore the same sports attire for each formal experiment. To avoid the effects of circadian rhythms on their physical state, experiments were conducted between 16:00 and 18:00. Identical equipment was used throughout the experiment, and verbal encouragement was consistently provided to the subjects.

**Table 2 Experimental intervention protocols.**

| Group | Load volume | Rest between sets |
|---|---|---|
| Control group | None | None |
| Experimental group A | 1 Set × 6 Reps | None |
| Experimental group B | 2 Sets × 6 Reps | 2 min |
| Experimental group C | 3 Sets × 6 Reps | 2 min |

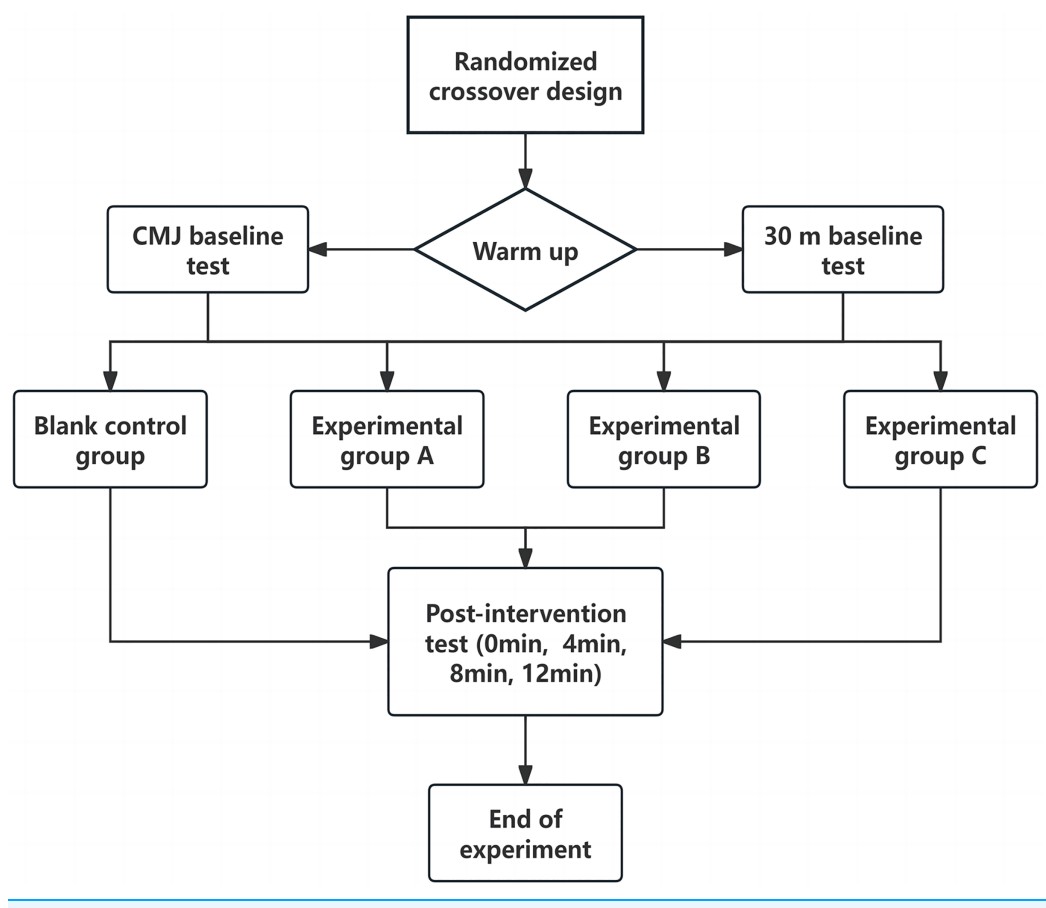

**Figure 2 Experimental procedure.**     

### Statistical analysis

All experimental data were recorded in Excel and then analyzed using SPSS 27.0 software (IBM Corp., Armonk, NY, USA). After testing for normal distribution, a two-factor repeated measures ANOVA (different intervention protocols × different recovery times post-intervention) was conducted, Mauchly sphericity test was conducted on the results. If the results were consistent with the sphericity test, the main effects of group and time on each index would be reported. In case of violation of sphericity, Greenhouse-Geisser correction would be applied to report the main effect, the Bonferonni method was employed for *post hoc* comparisons in the presence of main effects, with a significance level

**Table 3 Effects of different types of effects on CMJ height and 30 m sprint performance.**

|  | Effect type | F | df | P |
|---|---|---|---|---|
| CMJ height | Time main effect | 50.16 | 4 | <0.01 |
|  | Group main effect | 0.83 | 3 | 0.48 |
|  | Time × Group interaction effect | 5.27 | 12 | <0.01 |
| 30 m sprint performance | Time main effect | 44.02 | 4 | <0.01 |
|  | Group main effect | 0.14 | 3 | 0.94 |
|  | Time × Group interaction effect | 5.67 | 12 | <0.01 |

set at $P < 0.05$. To quantify the intervention effect, Cohen's d value was used to represent the ES, with ES < 0.2 considered trivial, $0.2 \leq ES < 0.5$ small, $0.5 \leq ES < 0.8$ moderate, and ≥0.8 large.

# RESULTS

## Effect of OPL flywheel half-squat intervention on CMJ height under different load volumes

The two-factor repeated measures ANOVA results showed that changes in CMJ height had a main effect of time (F = 50.16, df = 4, $P < 0.01$) and a time × group interaction effect (F = 5.27, df = 12, $P < 0.01$), but no main effect of group (F = 0.83, df = 3, $P = 0.48$). This indicates that CMJ height varied with recovery time post-intervention. At 0 min post-intervention, all experimental groups had CMJ heights lower than the baseline, with the degree of decline increasing with higher load volumes. At 4 min post-intervention, CMJ height began to rise in all experimental groups. Specifically, the CMJ height in experimental group A significantly increased during the recovery period of 4–8 min ($P < 0.01$); the CMJ height in experimental groups B and C significantly increased during the recovery period of 4–12 min ($P < 0.01$). The CMJ height in the control group which did not receive any intervention showed no significant difference from baseline at any recovery time ($P > 0.05$). The improvement in CMJ height peaked at 8 min of recovery for the flywheel half-squat schemes with different load volumes. However, in terms of ES, the improvement was Group B > Group C > Group A. Detailed results are presented in Tables 3 and 4 and Fig. 3.

## Effect of OPL flywheel half-squat intervention on 30 m sprint performance under different load volumes

Similar to the CMJ results, the two-factor repeated measures ANOVA showed that changes in 30 m sprint completion time had a main effect of time (F = 44.02, df = 4, $P < 0.01$) and a time × group interaction effect (F = 5.67, df = 12, $P < 0.01$), but no main effect of group (F = 0.14, df = 3, $P = 0.94$). This indicates that 30 m sprint performance varied with recovery time post-intervention. At 0 min post-intervention, all experimental groups had 30 m sprint completion time higher than the baseline, with the degree of increase rising with higher load volumes. At 4 min post-intervention, the 30 m sprint

**Table 4 Comparison of CMJ height at different recovery time points post-intervention with baseline.**

| Group | Time point | CMJ height (cm) | ES |
|---|---|---|---|
| Control group | Baseline | 47.82 ± 4.52 | —— |
| | 0 min | 47.84 ± 4.69 | 0.00 |
| | 4 min | 47.64 ± 4.39 | −0.04 |
| | 8 min | 47.49 ± 4.45 | −0.07 |
| | 12 min | 46.86 ± 4.33 | −0.22 |
| Experimental group A | Baseline | 47.91 ± 4.38 | —— |
| | 0 min | 47.36 ± 4.21 | −0.13 |
| | 4 min | 49.06 ± 4.3** | 0.27 |
| | 8 min | 49.38 ± 4.34** | 0.34 |
| | 12 min | 48.62 ± 4.16 | 0.17 |
| Experimental group B | Baseline | 47.67 ± 4.22 | —— |
| | 0 min | 46.97 ± 4.1 | −0.17 |
| | 4 min | 50.34 ± 4.15** | 0.64 |
| | 8 min | 51.34 ± 4.41** | 0.85 |
| | 12 min | 50.32 ± 4.5** | 0.61 |
| Experimental group C | Baseline | 47.79 ± 4.82 | —— |
| | 0 min | 46.45 ± 5* | −0.27 |
| | 4 min | 50.37 ± 4.89** | 0.53 |
| | 8 min | 51.63 ± 4.73** | 0.8 |
| | 12 min | 50.66 ± 5.11** | 0.58 |

**Note:**
In the figure, "*" indicates a significant difference compared to the baseline ($P < 0.05$), and "**" indicates a highly significant difference compared to the baseline ($P < 0.01$).

completion time began to decline in all experimental groups. Specifically, experimental group A significantly reduced the 30 m sprint completion time during the 4 min recovery period ($P < 0.05$); experimental groups B and C significantly reduced the 30 m sprint completion time during the 4–12 min recovery period ($P < 0.01$). The control group without intervention showed a significant increase in 30 m sprint completion time at 12 min compared to the baseline ($P < 0.05$), with no significant difference at other recovery times ($P > 0.05$). The improvement in 30 m sprint performance peaked at 4 min of recovery for experimental group A and at 8 min of recovery for experimental groups B and C. However, in terms of ES, the improvement was Group B > Group C > Group A. Detailed results are presented in Tables 3 and 5 and Fig. 4.

## DISCUSSION

The aim of inducing PAPE by preloading is to achieve better athletic performance during subsequent activities. Although the use of traditional resistance exercises to induce PAPE has been extensively studied, evidence supporting the use of flywheel for this purpose remains limited (*Beato & Dello Iacono, 2020*). This study aimed to explore the effects of OPL flywheel half-squat interventions with different load volumes on CMJ height and 30 m sprint performance in collegiate athletes. The experiments demonstrated that 1, 2,

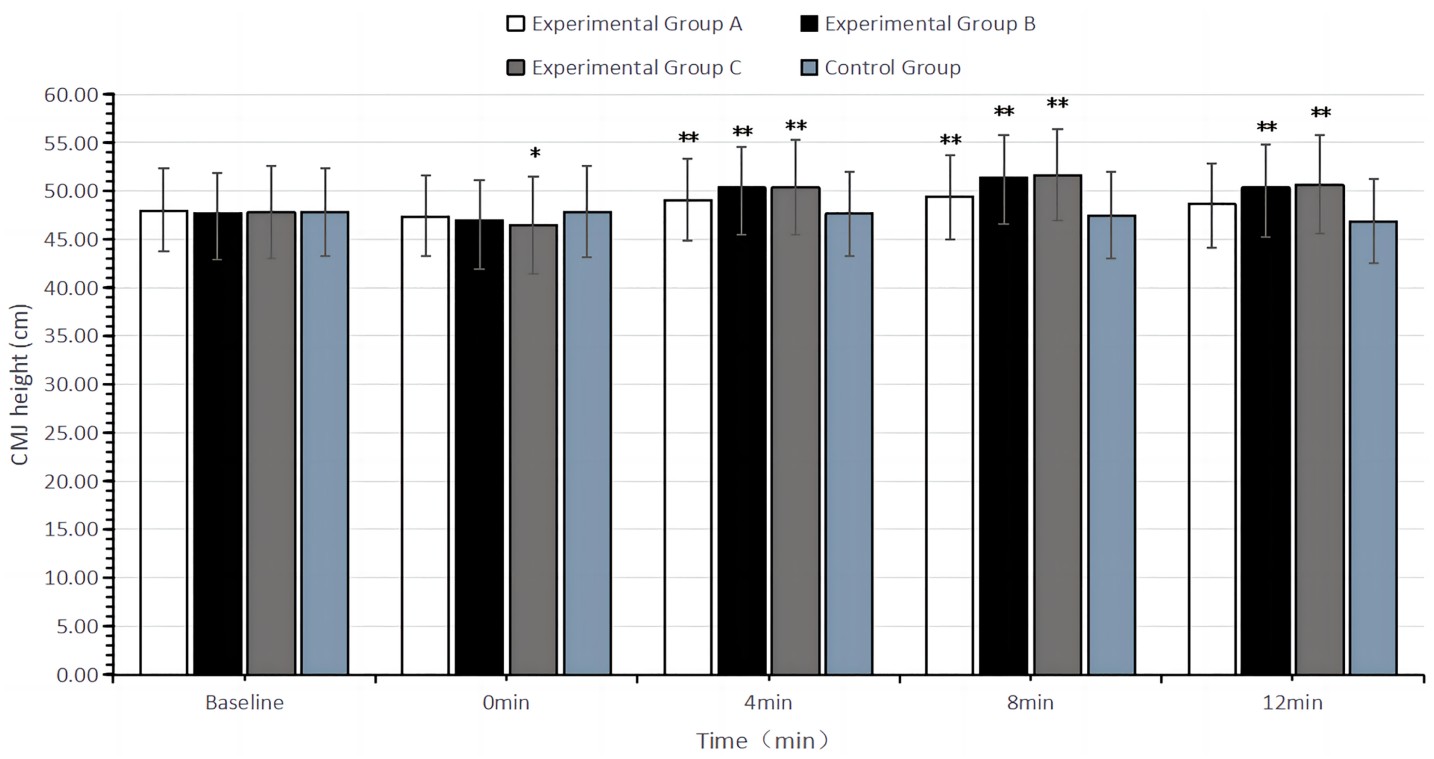

**Figure 3 Bar chart of CMJ height (cm) with different recovery times after intervention in different groups.** In the figure, "*" indicates a significant difference compared to the baseline ($P < 0.05$), and "**" indicates a highly significant difference compared to the baseline ($P < 0.01$).

**Table 5 Comparison of 30 m sprint performance at different recovery time points post-intervention with baseline.**

| Group | Time point | 30 m sprint performance (s) | ES |
|---|---|---|---|
| Control group | Baseline | 4.18 ± 0.10 | —— |
| | 0 min | 4.18 ± 0.11 | 0.01 |
| | 4 min | 4.19 ± 0.10 | 0.1 |
| | 8 min | 4.19 ± 0.11 | 0.15 |
| | 12 min | 4.20 ± 0.10* | 0.2 |
| Experimental group A | Baseline | 4.18 ± 0.11 | —— |
| | 0 min | 4.21 ± 0.11** | 0.29 |
| | 4 min | 4.16 ± 0.11* | −0.16 |
| | 8 min | 4.17 ± 0.10 | −0.1 |
| | 12 min | 4.19 ± 0.11 | 0.05 |
| Experimental group B | Baseline | 4.19 ± 0.11 | —— |
| | 0 min | 4.24 ± 0.14** | 0.33 |
| | 4 min | 4.14 ± 0.11** | −0.52 |
| | 8 min | 4.13 ± 0.10** | −0.64 |
| | 12 min | 4.17 ± 0.13** | −0.23 |

| Group | Time point | 30 m sprint performance (s) | ES |
|---|---|---|---|
| Experimental group C | Baseline | 4.19 ± 0.12 | —— |
| | 0 min | 4.24 ± 0.10** | 0.47 |
| | 4 min | 4.13 ± 0.12** | −0.52 |
| | 8 min | 4.12 ± 0.14** | −0.57 |
| | 12 min | 4.16 ± 0.10** | −0.23 |

Note:
In the figure, "*" indicates a significant difference compared to the baseline ($P < 0.05$), and "**" indicates a highly significant difference compared to the baseline ($P < 0.01$).

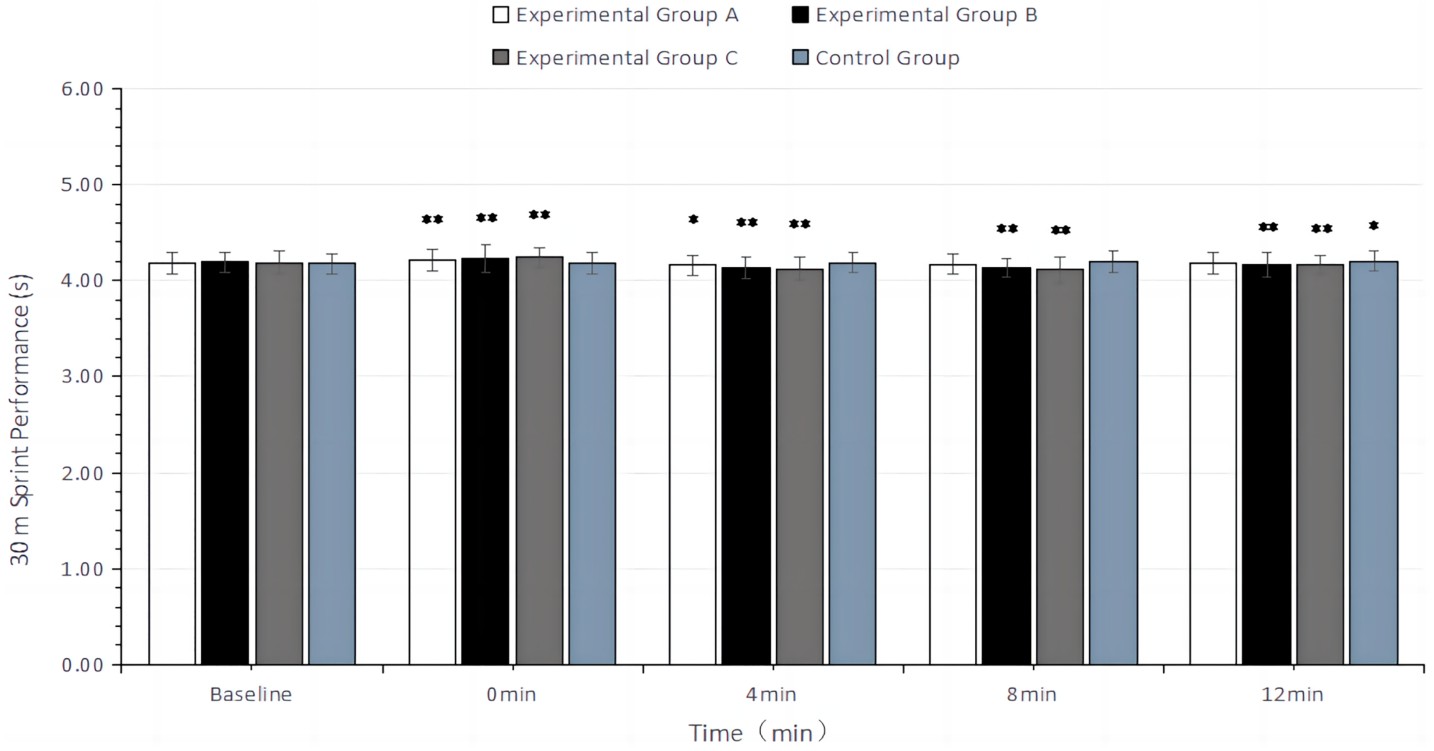

**Figure 4 Bar chart of 30 m sprint performance (s) with different recovery times after intervention in different groups.** In the figure, "*" indicates a significant difference compared to the baseline ($P < 0.05$), and "**" indicates a highly significant difference compared to the baseline ($P < 0.01$).

and 3 sets × 6 repetitions of flywheel half-squat interventions could induce significant PAPE after at least 4 min of recovery. However, considering the ES, the 2 sets × 6 repetitions flywheel half-squat intervention induced more significant PAPE compared to other training schemes (after 8 min of recovery). The reason for this result may be that too little or too much load volumes (1 sets × 6 reps or 3 sets × 6 reps) does not stimulate the body sufficiently or leads to a greater fatigue response. Some studies have shown that the triggering of PAPE by flywheel training may be related to the favorable neuromuscular responses and the high mechanical load induced during eccentric muscle actions, as well as

to a greater degree of selective recruitment of higher-order motor units during eccentric phases (*de Keijzer et al., 2020*; *Beato et al., 2019a*; *Beato & Dello Iacono, 2020*). However, the underlying physiological mechanisms need to be further elucidated (*Maroto-Izquierdo, Bautista & Martín Rivera, 2020*; *Beato & Dello Iacono, 2020*).

## Effects of different recovery times and load volumes on CMJ height

In competitive sports, jumping ability is crucial for achieving victory, such as in basketball for rebounding or in volleyball for spiking and blocking the net (*Suchomel, Nimphius & Stone, 2016*). Similarly, CMJ height is frequently used to assess lower limb vertical explosive power and serves as an indicator of lower limb neuromuscular performance, widely applied in various settings (*Claudino et al., 2017*).

This study found that all experimental groups began to show PAPE at 4 min post-intervention, peaking at 8 min and declining at 12 min. A meta-analysis by *Wilson et al. (2013)* indicated that recovery times of 7–10 min were superior to 3–7 min in terms of inducing a PAPE effect. The conditioning activities (CA) included dynamic lower body, static lower body, and dynamic upper body exercises. This study confirms that finding, showing that CMJ height peaked at 8 min post-intervention and remained elevated until at least 12 min. *Beato, Stiff & Coratella (2021)* found no increase in CMJ height at 15 s and 1 min of recovery after a flywheel half-squat intervention, but significant increases were observed at 3, 5, 7, and 9 min. Although their study did not use individualized loads (*e.g.*, OPL), it suggests that at least 3 min of recovery is required to observe significant increases in CMJ height post-intervention. Other studies have shown similar results, indicating that a recovery time of ≥3 min is necessary to observe improvement in CMJ height, whereas <3 min may not induce PAPE (*de Keijzer et al., 2020*; *Beato, Stiff & Coratella, 2021*; *Beato et al., 2019b*). The main factor likely contributing to this is that fatigue outweighs the enhancement effect at recovery times <3 min (*Beato, Stiff & Coratella, 2021*). It is noteworthy that this experiment found a decline in CMJ height improvement at 12 min, whereas *Maroto-Izquierdo, Bautista & Martín Rivera (2020)* observed a decline at 16 min. This discrepancy may be attributed to differences in the subject populations, the participants recruited in their study had at least 1 year of experience with flywheel training, whereas our participants were engaging in flywheel training for the first time. Therefore, this suggests that a greater familiarity with CA may also influence PAPE to some extent, although we provided flywheel training and conducted three familiarization sessions for the participants.

*Hamada et al. (2003)* found that load volume is a crucial factor affecting PAPE. Excessive load volume can lead to overfatigue, causing muscle fatigue to dominate during the PAPE peak period, whereas insufficient load volume fails to induce PAPE. Generally, athletes with extensive training experience can gain more enhancement benefits from higher training volumes, especially when using OPL (*Garbisu-Hualde & Santos-Concejero, 2021*). Therefore, this study compared the effects of flywheel half-squat interventions with different load volumes on CMJ height in collegiate athletes under individualized training loads (OPL). The results showed that at 0 min post-intervention, the decline in CMJ height was greater with higher load volumes. However, at the PAPE peak, the peak height was

positively correlated with load volume. In other words, the greater the load volume, the more significant the increase in CMJ height. Yet, the peak ES for each group indicated that the 2 sets × 6 repetitions flywheel half-squat intervention more reliably induced PAPE. This is similar to the findings of *de Keijzer et al. (2020)*, where at least two sets (six repetitions per set) were needed to induce significant PAPE using flywheel half-squats, with no significant difference between 2 and 3 sets. However, several studies using flywheel half-squats for PAPE induction have employed a 3 sets × 6 repetitions design (*Beato et al., 2021b*; *de Keijzer et al., 2020*; *Beato, Stiff & Coratella, 2021*; *Beato et al., 2019b*; *Xie et al., 2022*). While this can induce significant PAPE, this study and previous research suggest that 2 sets × 6 repetitions can induce similar PAPE magnitudes as 3 sets, reducing the fatigue caused by higher load volumes.

## Effects of different recovery times and load volumes on 30 m sprint performance

Short-distance sprinting ability is closely related to performance in many specialized sports and is frequently used in training practice as an important indicator of lower limb horizontal explosive power (*Xie et al., 2022*; *Fu et al., 2023*).

Similar to the results for CMJ height, all experimental groups showed increased sprint times compared to baseline at 0 min post-intervention. With extended recovery time, sprint performance began to improve at 4 min of recovery. At this time point, the improvement in sprint performance for experimental group A reached its peak, while groups B and C peaked at 8 min of recovery, followed by a decline at 12 min. The control group without intervention showed a significant increase in sprint completion time over baseline at the 12 min interval. This also indicates, to some extent, that the previous test time point did not produce an enhancing effect on the subsequent test time point during post-testing.

The reason for choosing 30 m as the test indicator in this study is based on *Beato et al. (2019b)*, who found that too short a sprint distance, such as 5 m, may reduce the reliability of experimental results. They suggested selecting longer distances (*e.g.*, ≥10 m) to improve reliability. However, this study found that the peak ES of the 30 m sprint in each experimental group was lower than the peak ES of CMJ height. This outcome can be explained biomechanically. Research suggests that PAPE is more likely to be maximized when the exercise inducing PAPE has similar biomechanical characteristics to the specific athletic task (*e.g.*, vertical or horizontal movements) (*Beato et al., 2019a*). Therefore, half-squats might not be the optimal exercise choice to induce PAPE for short-distance sprints. Instead, exercises with similar biomechanical features, such as hip thrusts or single-leg exercises, might be more appropriate.

In previous research, *Xie et al. (2022)* compared the effects of different inertias on 30 m sprint speed in basketball players and found that, compared to high inertia ($0.075$ kg·m$^2$), lower inertias ($0.015$ and $0.035$ kg·m$^2$) could induce PAPE for 30 m sprints at 9 min. Similarly, *Fu et al. (2023)* showed that medium inertia ($0.057$ kg·m$^2$) could induce PAPE for 30 m sprints at 4 min, compared to high inertia ($0.122$ kg·m$^2$). This might be because higher inertia produces more fatigue, interfering with the enhancement effect (*Beato et al.,*

*2021b*). The explanation given by the authors is that the 30 m sprint mainly involves rapid stretch-shortening cycle (SSC) activities. However, as inertia increases, the exercise speed during training slows down, potentially preventing the activation of the rapid SSC, which could partly explain why 30 m sprint performance does not improve after recovery (*Fu et al., 2023*). However, the load intensity in this study was determined by testing each subject's OPL. Research shows that OPL can maximize power output while balancing strength and speed, enhancing CMJ, standing long jump (SLJ), and change of direction (COD) performance (*Liang, Li & Niu, 2020*; *Freitas et al., 2019*). Therefore, using power to monitor flywheel load intensity might be more effective than other intensity monitoring methods (*Beato et al., 2024*). Additionally, due to the relatively small size and light weight of the flywheel training device, it can be used in more training environments and induce significant PAPE.

### Limitations and outlook

This study included only male collegiate athletes, excluding female or adolescent populations. Future research should further explore the acute effects of flywheel training on different groups. Additionally, the impact of various load volumes on PAPE should be investigated further, such as changes in PAPE with altered single-set repetition counts. Considering the positive correlation between biomechanics and PAPE (*Beato et al., 2019a*), future studies should explore how different flywheel exercise forms, which share similar biomechanical characteristics with specific athletic tasks, affect performance in those tasks. For instance, whether hip thrusts or single-leg exercises can improve short-distance sprint performance more significantly than half-squats. This study did not quantify the load of warm-up activities conducted prior to the baseline tests. Indeed, it must be acknowledged that the intensity of these warm-up activities could significantly affect the ultimate experimental results. Future studies should precisely quantify the intensity of warm-up activities and utilize objective indicators, such as heart rate, to provide detailed analysis. Furthermore, this study did not specify the interval between repeated CMJ tests conducted at the same time point (A 2–3 s interval is used). To enhance the reliability and validity of PAPE indicator test results, it is crucial to standardize the protocol, including how to schedule intervals for repeated testing of the same index at the same time point.

## CONCLUSIONS

Using 2 sets × 6 repetitions of OPL flywheel half-squat training induces a more reliable PAPE effect compared to higher load volumes. However, when using half-squats as the pre-stimulation exercise, the PAPE effect on CMJ height is greater than on 30 m sprint speed. It is recommended that strength and conditioning coaches and researchers adopt a 2 sets × 6 repetitions OPL flywheel half-squat protocol as a pre-competition warm-up strategy for tasks or sports that require maximal vertical displacement to achieve optimal PAPE response.

## Practical applications

The results of this study provide some practical application references for relevant practitioners, but it should be noted that any training program should be implemented according to the specific conditions of athletes. The most important finding of this study is that only 2 sets × 6 repetitions are required to maximize the PAPE effect after personalizing the inertial load intensity of the flywheel. Although most studies have proved that uniform inertia can also induce significant PAPE effect, we suggest that the load should be individualized according to the actual situation of each athlete in flywheel training to avoid overtraining or underload. In addition, athletes' OPL changes with long-term strength training. Coaches should regularly conduct assessments to determine the current OPL based on the athlete's strength level, in order to achieve optimal enhancement effects.

## ACKNOWLEDGEMENTS

Heartfelt thanks to all subjects who volunteered to participate in this study.

### Funding

The authors received no funding for this work.

### Competing Interests

The authors declare that they have no competing interests.

### Author Contributions

- Haonan Qi conceived and designed the experiments, performed the experiments, analyzed the data, prepared figures and/or tables, authored or reviewed drafts of the article, and approved the final draft.
- Mushuai Hao conceived and designed the experiments, performed the experiments, analyzed the data, prepared figures and/or tables, authored or reviewed drafts of the article, and approved the final draft.
- Boyang Qu analyzed the data, authored or reviewed drafts of the article, and approved the final draft.
- Liang Zhao analyzed the data, authored or reviewed drafts of the article, and approved the final draft.
- Wei Han conceived and designed the experiments, analyzed the data, authored or reviewed drafts of the article, and approved the final draft.

### Human Ethics

The following information was supplied relating to ethical approvals (*i.e.*, approving body and any reference numbers):

Shandong Sport University (Approval No.: 2022030) and complied with the ethical requirements of the Helsinki Declaration and relevant Chinese laws and regulations.

## Data Availability

The raw data are available in the Supplemental Files.

## Supplemental Information

Supplemental information for this article can be found online at http://dx.doi.org/10.7717/peerj.19321#supplemental-information.

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
