# Peer review of "Acute effects of optimal power load flywheel half-squat training on lower limb explosive power under different load volumes"

_PeerJ, doi:10.7717/peerj.19321_

## Round 0.1 · original submission · Major Revisions

Dear authors,

Manuscript titled "Acute effects of flywheel half-squat training with different load volumes on lower limb explosive power" that you submitted to PeerJ has been reviewed.

The reviewers have suggested that some important points must be clarified and have requested substantial changes to be made in the manuscript. Therefore, I invite you to respond to the reviewers' comments and revise your manuscript. The reviewers comments are included at the end of this letter.

Please ensure that all review, editorial, and staff comments are addressed in a response letter and that any edits or clarifications mentioned in the letter are also inserted into the revised manuscript where appropriate.

Reviewer 1 ·

Basic reporting

1. The phenomenon you are describing in the paper is not Post-activation potentiation (PAP) but Post-activation Performance enhancement (PAPE), which has been differentiated by previous research. These differ both in definition, as well as in underlying mechanisms (Xenofondos et al, 2023; Blazevich & Babault, 2019).

2. Lines 23-31. In this introductory paragraph, you should expand, with at least 2 concise sentences, on PAPE mechanisms and how PAPE is affected by the balance between fatigue and potentiation (Blazevich & Babault, 2019; Xenofondos et al, 2018) . This will set up your rationale and study justification for investigating volume with individualized loads and your discussion section.

3. There is a consistent reference to load volumes throughout. However, this is ambiguous.
You didn’t provide a definition for what you consider load volume, and considering the classical definition of volume load, you didn’t investigate different load volumes. However, you have investigated different volumes at individualized inertial loads. Investigating Load Volumes would entail setting pre-specified load-volumes that are equal for subjects within groups:
i.e. Group 1 = 0.24 Volume load, Group 2 = 0.40 Volume load … etc
Where for example, Volume Load would be measured:
Group A – pre-defined volume load = 0.24
Subject A - Optimal power load = 0.01 kg/m2
Subject B - Optimal power load = 0.02 kg/m2
Then, Group A intervention could be:
-For Subject A:
4x6@ inertial load 0.01 kg/m2 = 4x6x0.01 = 0.24 Volume load
-For Subject B:
4x3@ inertial load 0.02 kg/m2 = 4x3x0.02 = 0.24 Volume load
OR
2x6@ inertial load 0.02 kg/m2 = 4x6x0.01 = 0.24 Volume load
Group B – pre-defined volume load = 0.40
So on..
This is just an example check these studies to see about how others have defined volume load (Carvalho et al., 2022; Paoli et al., 2017; Stone et al., 2020).
Furthermore, the application of individualized inertial loads precedes a standardization method applied for subjects, and since this is not compared with a non-standardized inertial load intervention, the load is considered a constant throughout the study and is therefore not considered to be investigated. Therefore, it is suggested this study’s interpretation/reporting should be set to a) Different volumes with individualized loads, OR b) Different set configurations with individualized loads OR something of the same nature. This reporting should apply throughout the text and the title, giving the reader a clear understanding of what exactly is being investigated.
4. Raw dataset. The columns need identifiers. For dummy coded variables, consider a) providing an extra file with the description of the meaning of the dummy coded variables, or b) re-coding them with character definitions so that they are clear and understandable.

5. Line 39-43. This sentence is incorrect and should be removed. Traditional exercises do not determine the pause between eccentric and concentric movements. This is solely attributed to the execution-style/speed of the exercise, and squatting with a flywheel does not attenuate this issue. Furthermore, the citation reported here does not support the flywheel in this context.

6. Lines 59-61. Studies have included different loads or individual load intensities in their investigations. This should be stated. Different loads (Shi et al., 2024) and individualized loads (Maroto-Izquierdo et al., 2020; Sañudo et al., 2020).

7. Lines 61-62. Here, load and intensity are used together as “load intensity”, but in this context, they are synonymous. Keep only load, so as to remain consistent throughout the text.
Intensity, in your context, refers to maximal voluntary contractions with the intent of concentric contraction performed as quickly as possible, as you stated in lines 102-105.

8. Line 69. “PAP” is not needed here. Just state “acute effects of flywheel..” OR “PAPE effects of flywheel..”

9. The inclusion of some more information like means and standard deviations of
baseline measurements for CMJs, sprint times and OPL inertial loads in the demographics table may improve the reader's understanding of the population.

Experimental design

1. Lines 110-113. The meta-analysis cited does not mention PAPE warm-ups, please cite here the relevant studies. Although the employed warm-up may well be considered an adequate general warm-up, there is no mention of a specific warm-up for either jump or sprint tasks (Lines 140-150). This is a major limitation, as any improvements after the intervention may be attributed to additional high-intensity warm-up. This should be explicitly stated as a possible limitation. See pages 13-14 (Blazevich & Babault, 2019).

2. Lines 140-142. 2-3 seconds of rest is neither typical nor adequate to be used between jumps to determine peak jump height; furthermore, the cited study used three jumps with 1 minute of rest between jumps to determine peak jump height (Beato et al., 2021). If this was based on a different study’s protocol, please cite the study, otherwise this should be explicitly stated in the limitations.

3. Line 132-140. The bar's exact positioning and the linear sensor's placement are very confusing. Lines 132-135 It is stated that one end of the linear sensor is placed on the bar and the other *horizontally behind the subject's neck. Yet it is also stated they are also holding the bar while jumping (Lines 137-140)? Lines 137-138. Was the bar held in front of the body? Why would there be an arm swing, and how would grip the bar tight counter it?
Please provide a clearer explanation of the setup used to give the reader a clear understanding of how you measured the jump height. Especially since the setup used, may have slightly differed from the setup your noted citations used, as I understood, which are the ones that provide validity for the method. You may consider using a graphic to depict your setup clearly.

4. Lines 171-176. The type of tests used to conduct direct comparisons with baselines should be explicitly reported. Were multiple independent comparisons performed? Or were these a result of post-hoc analyses? The type of those tests should be specifically stated (i.e. Bonferonni, Tukey’s HSD..).

5. Lines 90-94. The time interval between each experiment/test session for each participant should be explicitly stated. Was this standardized? i.e. Participants participated in an experimental session every 5 days? 72hours? 96hours?

6. Line 105-107. What you have stated here is not clear and not in accordance with your noted citation. Where the review of Bright et al. 2023 states that, eccentric overload is not considered a given with flywheel training, and specifically states that: “To overcome this, authors have recommended that participants are instructed to free fall during the first third of the eccentric phase before applying a maximal effort to decelerate the rotating flywheel”. If this is the case, the sentence needs to be rephrased for better reader comprehension. The review also explicitly states that these specific instructions should also be used in the familiarization sessions. Please State here if you have used these instructions explicitly in your familiarization sessions.

Validity of the findings

The findings of this study offer meaningful insights towards the subject at hand. However, it is crucial that the limitations pointed out in the reporting and methodological sections are sorted and stated explicitly in the paper so that the paper offers the reader a complete picture of the circumstances that have led to the outcome of the study. Failing to do so, may lead to misinterpretation or non-replicability of the findings.

·

Basic reporting

This paper stands out positively when looking at the topic of sports sciences, performance, and athletes' health. However, I think some things need clarifying for the publication that will help in the overall interpretation and understanding of the results.

Abstract and Manuscript Structure
Comment 1 – The abstract should be structured as follows: Background, Methods, Results, and Conclusions. Additionally, the Authors should structure their manuscript according to the PeerJ guidelines: Introduction, Materials & Methods, Results, Discussion, Conclusions, Acknowledgements, and References.

Discussion and Conclusion
Comment 2 – Please add the Limitations and Strengths, and Practical Applications sections
References

Comment 3 – The references should be in Peer-J Style.

General Comment - Written English should be improved.

Experimental design

Comment 4 – The authors cited „......Helsinki Declaration..” Please add a reference.

Comment 5 - For the readers to better understand the protocol intervention, please add a timeline with specific intervention moments (training, assessments, ....).

Comment 6 - Are all the participants in this study athletes in the same sport? Identify the sports disciplines and competitive level of the participants.

Comment 7– The authors refer to ‘light resistance’. How much? 65% rm, 50%? Please indicate the specific load.

Validity of the findings

Comment 8 – The authors used 20 participants, but they didn't explain how they calculated the Power Sample. Please indicate how they carried out the calculations and when the outcome was for these clusters.

Comment 9 – The authors conducted a randomized study. However, they do not explain the technique used to randomize the study groups. Please add the sample randomization technique.

Reviewer 3 ·

Basic reporting

The manuscript investigates an intriguing aspect of exercise science, focusing on the acute effects of flywheel half-squat training with different load volumes on lower limb explosive power. The study is timely and adds to the growing body of literature on BFRT, an area of keen interest due to its implications for both athletic performance and rehabilitation. However, to meet the high standards of a prestigious journal, several areas require significant attention and improvement.

Experimental design

The study utilizes a randomized self-controlled crossover design, which is appropriate for the research question. However, the manuscript should provide more detailed justifications for the chosen sample size (n=20). Is this number sufficient to achieve statistical power, and were any power calculations conducted beforehand?

Validity of the findings

The study provides valuable insights into the acute effects of flywheel half-squat training with different load volumes on lower limb explosive power. The manuscript requires some revisions to enhance its clarity, methodological rigor, and the depth of its analytical discussions.

Additional comments

1.The manuscript is well structured and clearly articulates the research hypothesis and objectives in the introduction. It’s crucial to contextualize the study within the existing literature and explicitly state the research gap it intends to fill.
2. Future research directions proposed by the study are valuable. The manuscript should add and discuss potential physiological mechanisms underlying the observed effects and consider any counterintuitive or unexpected findings.

Annotated reviews are not available for download in order to protect the identity of reviewers who chose to remain anonymous.

---

## Round 0.2 · Minor Revisions

Dear authors,

The study entitled “Acute effects of optimal power load flywheel half-squat training on lower limb explosive power under different load volumes” demonstrated interesting findings using an appropriate methodological approach. However, minor revisions must be clarified in the manuscript. Your article has great potential for publication on PeerJ, but the reviewer has requested additional changes to be made.

Reviewer 1 ·

Basic reporting

The authors have made a great job at the revisions mentioned in the initial review. The revised manuscript now offers more detail and clarity, clearly articulating the significance and importance of the study’s findings in relevant contexts.
However, some revisions regarding the explicit mentioning of some limitations may have been overlooked. If these should be addressed, then the manuscript will be considered complete.

Experimental design

No comments

Validity of the findings

Validity of findings
In lines 437-442 of Tracked changes file and lines 398-402 of the pdf manuscript. The limitations stated here are vague and obscure and not sufficiently detailed. The limitations should be expressed with sufficient detail for readers and other researchers to understand the exact implications of the mentioned details on the inferences made by the authors and provide nuance in the interpretation of results.
In lines 304-306 of Tracked changes file and lines 281-283 of the pdf manuscript. Here the results of the study state that the control group experienced an enhancement in sprinting at 12 mins after the warm up. This is significant and should not be overlooked as this could be a result of two plausible scenarios. It is possible that this could be a result of the learning effect, or a result, of additional high intensity specific warm-up performed after the general warm-up conducted in the study. Where additional 30m sprints at 0mins, 4mins, 8mins, conducted after the general warm-up may have constituted as high intensity sprint specific warm-up in this case, thus enhancing performance at 12mins for the control group. Either way, this has a significant impact on the validity of the sprint findings, because it could also be extrapolated to inferring that the inclusion of the flywheel protocol in the other groups may have acted as an additional high intensity warm-up. Furthermore, this could also be elaborated to affect the validity of the CMJ findings.
Therefore, the authors must explicitly mention:
a) In their discussion’s section, they should include a detailed but concise discussion of the possible reasons that the control group had a significant increase in performance for sprint at 12 minutes and how this may affect the inferences made by the authors (most plausible of which are mentioned above).
b) In the limitations section, that their warm-up strategy did not include any high intensity specific warm-up beyond their general warm-up (5mins of slow running and 5mins of dynamic exercises), before conducting baseline tests.
c) In their limitations section, that their rest period between the baseline jumps which consisted of 2-3 seconds, could also be a potential limitation.
While we thank the authors for making substantial changes to the manuscript that we previously suggested, these changes should still be applied. These will provide clarity of the possible factors influencing this study’s results and will give the reader or researcher the appropriate information to judge the study’s findings independently and make their own inferences.

---

## Round 0.3 · Minor Revisions

Dear authors,

The study entitled “Acute effects of optimal power load flywheel half-squat training on lower limb explosive power under different load volumes” demonstrated interesting findings using an appropriate methodological approach. However, minor revisions must be clarified in the manuscript. Your article has potential for publication on PeerJ, but the reviewers have requested substantial changes to be made.

·

Basic reporting

The authors haven't replied to my comments.

Experimental design

The manuscript continues to contain methodological errors, with procedures that are not explained (i.e.: body composition analysis, randomization of sample, .....).

Validity of the findings

None to reported.

Additional comments

The manuscript continues to contain methodological errors, with procedures that are not explained (i.e.: body composition analysis, randomization of sample, .....). For these facts, my suggestion is to reject the manuscript.

Reviewer 3 ·

Basic reporting

no comment

Experimental design

no comment

Validity of the findings

no comment

Additional comments

no comment

Reviewer 4 ·

Basic reporting

Overall, the article is well-written, although some considerations need to be taken into account.

Abstract
Line 43-46: PAPE itself refers to enhancement; consider revising lines 43–46.

Introduction
Line 68: The conditioning activity (CA) used to induce the PAPE effect can also be of high intensity (Brink et al., 2022). Consider making the necessary changes.
Line 77: PAPE is a response (i.e., an enhancement in performance) resulting from the CA; therefore, the following line requires revision: "PAPE can be utilized as a warm-up strategy to enhance subsequent athletic performance."

Experimental design

The experimental design is clear.

Validity of the findings

The findings presented in the manuscript are clear and supported by the data; however, there are some minor issues that should be addressed.

Line 278-280: Given that PAPE is a response (i.e., an enhancement in performance) resulting from the CA, the phrase “PAPE of the experimental group” sounds ambiguous. Suggest revising it.
Line 279-280: The authors used the term “sprint speed,” although sprint completion time was measured. The authors stated “significantly decreased 30m sprint speed.” Does this mean the completion time increased (i.e., a decrement in performance) for the experimental group? However, the results in the table suggest an enhancement in performance (i.e., a reduced completion time).
Line 281-283: Use either “sprint speed” or “sprint completion time.” To avoid confusion, the authors could consider using the term “sprint performance” and defining it as “sprint completion time”.
Line 322-324: Please indicate the CA reported in the meta-analysis.
Line 324: “CMJ height PAPE peaked”. Consider using just “CMJ height peaked”.
Line 326: To represent time, either seconds (s) or minutes can be used in the manuscript, but consistency should be maintained. For example, "min" is used in some instances (Lines 260, 264, 282, etc.), while "minutes" is used in others (Line 322, etc.). Please check for consistency throughout the manuscript.
Line 330: “CMJ height PAPE” - consider using “enhancement/improvement in CMJ height”.
Line 332: Consider using an alternative term for “potentiation” to avoid confusion between PAP and PAPE.
Line 335: Did the study by Maroto-Izquierdo et al. (2020) use flywheel/OPL? Additionally, is the difference in the subject population attributed to factors such as sports or training experience? Consider expanding on this.
Line 340: Consider using an alternative term for “potentiation” to avoid confusion between PAP and PAPE. Same for Line 382.
Line 355: Consider using the terms “sprint performance” or “sprint completion time.” Similarly, review Lines 360 and 386 for consistency. Please ensure this is checked throughout the manuscript.
Line 364: Do you mean “at 12 min time interval”? Suggest revising it.
Line 369: Did the authors also measure the reliability of the performance tests used in this study?
Line 388: Please abbreviate LJ and COD.
Line 398-399: Do you mean exercises that share similar biomechanical characteristics with the athletic task? If so, consider revising the line for clarity. Additionally, please provide a relevant reference to support the claim.
Line 416: Consider using an alternative term for “induction”. For example, can use “response”.
Line 425-426: Please revise, as it is unclear.

References:
Brink, N. J., Constantinou, D., & Torres, G. (2022). Postactivation performance enhancement (PAPE) of sprint acceleration performance. European journal of sport science, 22(9), 1411–1417. https://doi.org/10.1080/17461391.2021.1955012

Additional comments

The references presented in the list are in numerical order, rather than the alphabetical order used for citations in the manuscript.

Reviewer 5 ·

Basic reporting

Specific comments:

Line 139-140: Please include the specific resistance training experience of participants with inertial flywheel training.

Line 210: Please include more details of VBT Pro, ie, manufactured by who and where.

Line 347-348: “Wilson et al. (2013) indicated that recovery times of 7-10 minutes were superior to 3-7 minutes”. Superior for what? Please include to induce a PAP effect or similar.

Line 419: The abbreviation L J is used without prior explanation, please include long jump (LJ) or the desired meaning.

Line 449: “It is recommended that strength and conditioning coaches and researchers adopt a 2 sets × 6 repetitions OPL flywheel half-squat protocol as a pre-competition warm-up strategy to achieve optimal PAPE induction”.
Suggest including: as a pre-competition warm-up strategy for tasks or sports that require maximal vertical displacement.
As the results of the current study and discussion highlight the importance of conditioning contraction specificity (i.e. half squat employed for CMJ, and the potential of hip thrust to be beneficial for sprinting) this needs to be specified here.

Line 456: Consider replacing “good” with significant.

Experimental design

A well designed random crossover design, the authors are commended for the thought put into the study design.

Validity of the findings

no comment

Additional comments

The authors are commended for the thought process behind this study. A well-structured random crossover design that highlights an interesting gap in the PAPE and inertial flywheel training literature. Where the use of optimal power load (OPL) has been utilised to assess its effect on both acute CMJ and sprint performance with varying loading volumes.
The authors have referenced and built on the previous research well. This reviewer has minimal comments to be addressed. The manuscript is very well written and adds significantly to the PAPE and flywheel training literature. All tables and figures are clear and easy to interpret while providing the reader with the relevant information.
See below a few minor comments which will hopefully improve the quality of the manuscript in the future.

Annotated reviews are not available for download in order to protect the identity of reviewers who chose to remain anonymous.

---

## Round 0.4 · Minor Revisions

Dear authors,

Manuscript titled "Acute effects of optimal power load flywheel half-squat training on lower limb explosive power under different load volumes" that you submitted to PeerJ has been reviewed.

The authors did a good good work of revisions in response to the reviewers' comments. However, minor revisions must be clarified in the manuscript. The reviewer(s) comments are included at the end of this letter.

Reviewer 4 ·

Basic reporting

Good work on addressing the suggestions. However, there are a few very minor issues that should still be considered.

Abstract
Line 45: Consider using the term “sprint performance” or “sprint completion time” instead of “sprint speed”. Please check this in the tables and figures as well.

Introduction
Line 71: replace “height” with “high”.

Experimental design

Line 229: Replace “30m” with “30 m”. Check this throughout the manuscript.

Validity of the findings

The findings presented in the manuscript are clear and supported by the data.

Line 334: Consider using “30 m sprint performance” instead of “30 m PAPE”.
Line 340: Add a space after “x”.
Line 438: Abbreviation requires correction; consider: standing long jump (SLJ) and change of direction (COD).
Line 470: Remove the full stop after “displacement”.
Line 475: Add a space after “x”.
Line 482: Consider removing “when inducing PAPE”.

Reviewer 5 ·

Basic reporting

All previous reviewer comments have been addressed appropriately

Experimental design

All previous reviewer comments have been addressed appropriately

Validity of the findings

All previous reviewer comments have been addressed appropriately

Additional comments

All previous reviewer comments have been addressed appropriately

---

## Round 0.5 · accepted · Accept

Dear Author,

Congratulations, after the good work of revisions in response to the reviewers' comments, I would like to inform you that your manuscript has been accepted for publication in PeerJ.